# A Comparative Study between 18F-FDG PET/CT and Conventional Imaging in the Evaluation of Progressive Disease and Recurrence in Ovarian Carcinoma

**DOI:** 10.3390/healthcare9060666

**Published:** 2021-06-03

**Authors:** George Rusu, Patriciu Achimaș-Cadariu, Andra Piciu, Simona Sorana Căinap, Călin Căinap, Doina Piciu

**Affiliations:** 1Iuliu Hațieganu University of Medicine and Pharmacy, 400012 Cluj-Napoca, Romania; george.rusu@iocn.ro (G.R.); doina.piciu@umfcluj.ro (D.P.); 2Ion Chiricuță Institute of Oncology, 400015 Cluj-Napoca, Romania; pachimas@umfcluj.ro (P.A.-C.); calincainap2015@gmail.com (C.C.); 3Department of Surgical Oncology, Iuliu Hațieganu University of Medicine and Pharmacy, 400012 Cluj-Napoca, Romania; 4Department of Medical Oncology, Iuliu Hațieganu University of Medicine and Pharmacy, 400012 Cluj-Napoca, Romania; 5Department of Mother and Child, Iuliu Hatieganu University of Medicine and Pharmacy, 400012 Cluj-Napoca, Romania; simona.cainap@yahoo.com

**Keywords:** PET/CT, ovarian cancer, treatment management

## Abstract

The aim of this study is to compare the efficiency of conventional imaging and 18F-FDG PET-CT in detecting progressive disease and recurrences over a period of one year (2018), in the case of ovarian cancer, and also to assess the importance of 18F-FDG PET/CT in changing the course of the treatment for these patients. This study included 29 patients diagnosed in various stages with ovarian carcinoma, most of them of epithelial origin. All patients were evaluated throughout their treatment using 18F-FDG PET/CT and various conventional techniques (computed tomography (CT), magnetic resonance imaging (MRI), abdominal and intravaginal ultrasound, chest X-ray). PET/CT was more useful and effective in our group of patients in detecting progressive disease compared with conventional imaging (37.93% vs. 17.24%) and also in establishing the recurrences (24.14% vs. 6.90%). Moreover, F18-FDG PET-CT led to a therapeutic change in 55.17% of the patients of our group, compared with only 17.24% after conventional imaging. This underlines the crucial aspect of the metabolic changes of tumors that should be assessed alongside the morphological ones, with PET-CT imaging remaining the only viable tool for achieving that at present. PET/CT with 18F-FDG represents one of the most important imaging techniques used in the diagnosis and management of ovarian carcinoma. Our results seem to fall in line with what other authors reported, indicating that 18F-FDG PET-CT is potentially gaining more ground in the management of ovarian carcinoma, by influencing therapeutic strategies and by being able to detect relapse and progression accurately.

## 1. Introduction

Ovarian cancer is one of the most frequent gynecologic malignancies, being an important localization when taking into consideration both the prevalence and the mortality associated with this disease. In 2018, ovarian carcinoma was the seventh most prevalent malignancy among females worldwide, representing 3.3% of all cancer cases [1]. It is also responsible for 4.4% of cancer-related deaths among females [2]. In 2018, an approximate number of 295.414 new cases of ovarian carcinoma were reported [3], and in 2040, it is estimated this number will increase to 434.184 new cases [4]. Several risk factors have been associated with the development of this disease [5,6], such as the following: nulliparity; early menarche; late menopause; substitutive hormonal therapies during menopause; familial history of ovarian carcinoma. From a histological standpoint, ovarian tumors were previously classified as epithelial (derived from the surface coelomic epithelium) or non-epithelial (stromal or germinal tumors). Recent classifications allow, however, the separation of epithelial ovary carcinomas into low-grade and high-grade subgroups. The high-grade tumors seem to be the most prevalent worldwide. More rarely, there are described cases of stromal and germinal tumors [7]. Most cases are usually diagnosed as high-grade serous carcinomas, with most of them being in stage IIIc [8].

Regarding the metastatic spread of ovarian carcinoma, three potential routes have been described. The most common one is through the peritoneum, due to the normal peritoneal fluid circulation and distribution. Lymphatic spread is also a potential path for metastatic involvement, and, finally, less frequently, tumor dissemination may happen through the bloodstream, with this path being responsible for distant organ metastases.

Treatment in ovarian carcinoma is mainly surgical, through primary debulking, followed by chemotherapy/immunotherapy. Clinically advanced cases, where cytoreduction is deemed improbable through surgical means, benefit from neoadjuvant chemotherapy [9].

Routine screening for ovarian carcinoma is not currently recommended, despite the low rate of early diagnosis. As such, asymptomatic women do not usually undergo screening for ovarian carcinoma [9,10,11,12]. However, several prognostic scores, such as ROMA and RMI, have been used to identify the risk of ovarian masses to be of malignant origin, with similar results [13,14]. Transvaginal ultrasound or abdominal contrast-enhanced CT scans are usually the first imaging tools used in the diagnosis of ovarian cancer. The presence of cystic pelvic masses, especially with papillary and septum structures, along with ascites and CA-125 elevation, supports the diagnosis of ovarian carcinoma. As such, imaging techniques, including both conventional and nuclear medicine ones, are not commonly used for the screening of ovarian cancer. PET/CT with 18F-FDG represents, however, a commonly used tool in the management of this disease, providing valuable information alongside conventional techniques in various aspects of the pathology [15,16].

Preoperative assessment is usually conducted with the help of conventional imaging, particularly contrast-enhanced computer tomography. While F18-FDG PET-CT has a limited role in assessment of the primary tumor, it is a particularly useful investigation for determining the lymph node and distant metastases, especially when it comes to extra-abdominal spread [16,17]. F18-FDG PET-CT is also useful in the characterization of adnexal masses, specifically in differentiating benign from malignant structures, with a worse performance in differentiating between benign and borderline ones [18].

18F-FDG-PET/CT imaging was also proven a valuable tool in assessing the prognosis of ovarian cancer, through factors such as metabolic tumor volume, total lesion glycolysis and preoperative SUVmax being relevant in predicting the outcome of the disease or guiding the primary therapeutic approach during neoadjuvant chemotherapy [19,20,21].

Its usefulness comes into place also in relation to disease recurrence, where PET/CT seems to be superior to both conventional imaging (computed tomography (CT), magnetic resonance imaging abdominal and intravaginal ultrasound, chest X-ray) and the CA-125 assay in regard to both sensitivity and specificity, both in high-grade and low-grade carcinomas [22,23].

18F-FDG PET/CT can also be applied to determine the efficiency of the treatment, often being a decisive factor in deciding the opportunity for treatment changes. F18-FDG PET-CT is able to identify patients with a poor response to chemotherapy who would benefit from switching to the next line of treatment [24,25].

This study aimed to assess the impact of 18F-FDG PET-CT in the management of these patients, specifically in their therapeutic approach. For that, we evaluated the ability of PET-CT imaging in detecting relapse and progressive disease, when compared with conventional imaging (mainly CT and abdominal/transvaginal ultrasound).

## 2. Materials and Methods

### 2.1. Patients

This study included 29 patients diagnosed with various histological types of ovarian carcinoma, in different stages of the disease. The age of the patients was between 42 and 78 years old, with a mean age ± standard deviation (SD) of 58.06 ± 7.97 years. The patients were monitored throughout their treatment at the “Prof. Dr. Ion Chiricuţă” Institute of Oncology. All patients included in the study signed the institutional informed consent for using their data both for the diagnosis and treatment and for scientific purposes. All of the patients were evaluated in 2018 with both conventional and 18F-FDG PET-CT imaging techniques. All patients were evaluated through CT scans, ultrasound and 18F-FDG PET-CT. Only two of the patients performed MRI prior to PET-CT imaging. All of the patients had epithelial tumors, with the exception of one who was diagnosed with a yolk sac tumor. CA-125 was measured prior to imaging, with a normal laboratory reference value of <35 U/mL. In the table below, we provide the CA-125 value measured before both types of investigations, with the exception of one patient who did not had any recent measurements of the tumor marker in our institution.

CA-125 was measured with electrochemiluminescence techniques (ECLIA). The CA-125 measurement was performed before administering each of the chemotherapy cycles and also during each follow-up control. The normal value of the laboratory is considered <35 U/mL, with a limit of detection of 0.6 U/mL [13]. Venous blood was sampled, with a minimum volume of 0.5 mL of blood per sample. Sampling was performed under fasting conditions.

For most patients, the usual monitoring of the disease included the measurement of CA-125 serum values, pelvic and abdominal ultrasonography and chest X-ray, along with the clinical exam.

Pelvic and abdominal ultrasound was performed at each control during follow-up and monitoring, along with the physical exam. Ultrasound was also performed whenever an increase in the serum level of CA-125 was spotted. Suspect structures discovered on ultrasound imaging were afterwards investigated further through CT scans. Likewise, negative ultrasound examinations, in the case of CA-125 elevation, were also investigated further through CT.

Negative CT scans were evaluated afterwards through PET-CT with F18-FDG. As such, in trying to compare the efficiency of conventional imaging for the same value of CA-125, most of the conventional techniques during that time reference interval were CT scans, along with negative ultrasound and chest X-ray images.

Regarding PET-CT scans, patients placed in follow-up were examined after at least 2 months since the last chemotherapy cycle. Those that underwent the PET-CT examination between chemotherapy cycles were scheduled at least 2 weeks apart since the last one or immediately before the new one. As such, we tried reducing the “flare” effect to a minimum, removing an important factor that might cause false results.

From a histopathologic standpoint, 23 of the patients (79.31%) were diagnosed in stage III of the disease, 2 of them in stage IV (6.89%), 1 in stage II (3.44%) and 1 in stage I (3.44%). We also did not have at our disposal the full histopathological result in the case of 2 patients. The characteristics of the studied group are presented in Table 1.

### 2.2. Positron Emission Tomography/Computed Tomography (PET/CT) Imaging

The imaging for our study was performed with a GE Optima 560 PET/CT device. The patients were asked to fast for 6 h before the F18-FDG PET/CT studies were acquired. The level of blood glucose was carefully monitored prior to injecting the radiopharmaceutical, with a desirable interval of 70–150 mg/dL. The administered activity of the radiopharmaceutical varied between 185 and 600 MBq, being calculated according to the patient weight and the institutional protocol for tumor imaging. The imaging acquisition was performed at 60–90 min afterwards. Our CT images were acquired with a slice thickness of 3.75 mm and through a low-dose protocol (100–120 Kv, 50–100 auto mA, index noise of 20%). All of the PET/CT images were evaluated by a team, formed by a nuclear medicine physician and a radiologist. For all PET/CT studies, we used SUVlbm (the standardized uptake value lean body mass) as a semi-quantitative parameter for the F18-FDG uptake calculation, maintaining a standard protocol on the workstation (Volumetrix for PET-CT) [26].

### 2.3. Statistical Analysis

Statistical analysis was performed using GraphPad Prism 6.0 software. We calculated means, standard deviations, sensitivity and specificity, as well as the Pearson correlation coefficient. The interpretation of the Pearson correlation was as follows: r = 0.3–0.5 (low correlation), r = 0.51–0.7 (moderate correlation), r = 0.71–0.9 (high correlation) and r > 0.91 (very high correlation).

## 3. Results

We compared conventional imaging and PET/CT scans for the patients included in this study, assessing how often the two approaches led to a therapeutic change for these patients. We also evaluated in what percent of the cases the imaging techniques were able to detect disease progression and recurrence. The treatment response for solid tumors can be assessed through both conventional imaging and PET-CT with F-18 FDG. For the first, the 2009 RECIST criteria are commonly used [27]. In our analysis, we defined progressive disease identified through conventional imaging, using the RECIST criteria, as: an increase by 20% in the sum of the diameters of the target lesions, taking into account the smallest sum of those diameters during the monitoring of the patient or the finding of new lesions. In the case of non-measurable disease, for example, advanced peritoneal carcinomatosis, unequivocal evolvement can be used as a criterion for progressive disease. PET-CT can also be used to assess progressive disease according to PERCIST 1.1 criteria [28], especially in the case of new lesions, either by a positive scan after a negative baseline one or a positive PET-CT scan, without any prior baseline one, but with confirmation from CT (either before or after the PET-CT imaging study). For our study, we used the aforementioned criteria systems in evaluating the presence of disease progression, either from an anatomical or metabolic point of view. Out of the 29 cases, PET/CT was able to identify the presence of progressive disease in 11 of them (37.93%), compared with conventional imaging, which detected disease progression in only 5 cases (17.24%), Figure 1.

In our group, conventional imaging was able to detect progressive disease in five of the patients (17.24%). Three of them were confirmed as positive for progressive disease through CT imaging (10.34%), one through abdominal ultrasound and one through MRI, while PET-CT detected progressive disease in the case of 11 patients (37.93%). As such, PET-CT with F18-FDG was able to identify 20.69% more cases of progressive disease than conventional imaging.

It is important to mention the case in which progressive disease was detected with the help of abdominal ultrasound. According to the RECIST criteria, this imaging method is not typically recommended for the assessment of treatment response, usually requiring confirmation through MRI/CT. Prior to the ultrasound, the patient underwent two CT scans that did not suggest progressive disease. However, a quick and persistent elevation of the CA-125 level (up to 537 U/mL at the moment of the last CT scan), along with the detection of a new peritoneal lesion at ultrasound, led to a swift therapeutic change, thus the case was considered progressive disease. PET-CT later confirmed the presence of the lesion; however, most of the other ones were in remission at the moment of the evaluation, in comparison with the prior examination.

F18-FDG PET-CT was able to detect peritoneal progression in eight of the patients, lymph node extension in eight (out of which five were subdiaphragmatic and three of them supradiaphragmatic, including supraclavicular, latero-cervical, mediastinal and internal mammary lymph nodes) and hepatic extension in four, and the spleen was involved in two cases (Table 2).

Regarding progressive disease, conventional imaging had a sensitivity of 45.4% and a specificity of 100% in detecting it in relation to F18-FDG PET-CT.

There was no correlation, however, in our group between positive F18-FDG PET-CT scans and the CA-125 level (r = 0.072) or the FIGO grade (r = 0.06).

Conventional imaging detected recurrence of the disease in 6.90% of the cases (two patients), while PET/CT achieved that in 24.14% of them (six patients). CT scans only detected 4% of these recurrences. As such, conventional imaging missed the diagnosis of 17.24% of the recurrences, which were later identified through PET-CT, Figure 2. When related to F18-FDG PET-CT in the diagnosis of ovarian carcinoma recurrence, in our patient group, CT scans had a sensitivity of 16.6% and a specificity of 100%. There were not any correlations found between recurrence confirmed by F18-FDG PET-CT and CA-125 levels (r = −0.008) or the FIGO grade (r = 0.09). Recurrences confirmed by conventional techniques were also not correlated with CA-125 values (r = 0.26), but there was a low positive correlation between those confirmed by CT scans and CA-125 (r = 0.47).

The therapeutic approach was also modified in relation to the results of the various imaging techniques, with the treatment course being changed in the case of 16 (55.17%) of the patients after PET-CT and 5 (17.24%) of them after conventional imaging, Figure 3. 

In Figure 4 and Figure 5 we display two cases where, based on PET/CT scans, the therapies had major changings. The modified treatment strategies occurred in our group, are presented in Table 3.

## 4. Discussion

PET/CT with 18F-FDG represents one of the most important imaging techniques used in the diagnosis and management of ovarian carcinoma. The ability to evaluate a tumor and the potential metastatic lesions from a metabolic standpoint allows for earlier diagnosis and better assessment of the prognosis and therapy efficiency.

In our studied group, PET-CT with 18F-FDG was more useful than conventional imaging in detecting both relapse of ovarian carcinoma and progressive disease. Peritoneal spread is usually the main route of disease progression, the majority of the patients being diagnosed in the third stage of the FIGO classification. A total of 72.7% of the patients with progressive disease presented peritoneal involvement, confirming this pattern. However, the lymphatic pattern of spread was also seen in an equal number of patients examined through PET-CT imaging in our study. PET-CT has also proven itself in our studied group to be more effective than conventional imaging in detecting distant, extra-abdominal metastases, particularly in the case of supradiaphragmatic lymph node involvement. Nam et al. also reported that PET-CT was able to identify 15.8% of cases of unexpected extra-abdominal lymph node extension and 3.8% of cases with other synchronous tumors [29]. A meta-analysis published in 2012 by Yuan et al. [30], including 882 patients, revealed a better sensitivity for PET-CT in the detection of lymph node metastases (73.2%) when compared with MRI (54.7%) and CT (42.6%).

Almost half (45.45%) of the patients with progressive disease also presented hepatic or splenic involvement; thus, the bloodstream pathway of spread is also very important to assess. The ability of PET-CT to detect abdominal and extra-abdominal disease involvement is perhaps another argument in favor of its usage in ovarian carcinoma.

PET-CT with F18-FDG is a valuable tool in detecting relapse (24.14% of the patients compared with 6.90% in conventional imaging), but also disease progression (37.93% vs. 17.24%). F18-FDG PET-CT proves itself as a valuable tool for monitoring patients with treated ovarian carcinoma, especially when the clinical exam or the increase in the CA-125 level might suggest recurrence/progression, but the conventional imaging techniques remain negative.

A 2008 study [31] performed by Sebastian et al. showed a statistically significant difference in the accuracy of detection for ovarian carcinoma recurrences, when comparing CT with PET-CT, with the latter method being more efficient. The interobserver variability of results was also lower in the case of PET-CT. Additionally, Coakley et al. [32] described an 85–93% sensitivity for detection of peritoneal metastases in ovarian carcinoma, through spiral CT. However, the sensitivity was significantly lower for peritoneal implants smaller than 1 cm.

Antunovic et al., also suggested the superiority of PET-CT over conventional imaging techniques and CA-125 alone in detecting epithelial ovarian cancer recurrence [22] (80% accuracy vs. 62% vs. 64%), with the additional benefit of the PET-CT findings not being influenced by the histology of the tumor.

Muñoz et al. [33] also observed that PET-CT with 18F-FDG was more accurate than CA-125 alone in diagnosing early ovarian carcinoma recurrence (CA-125 was negative in 50% of the cases that were confirmed through PET-CT). Negative levels of CA-125 cannot exclude active disease, either as recurrence, residual disease or progression. Palomar et al. [34] also identified 53% of cases where CA-125 was below 30 UI/mL but had positive PET-CT 18F-FDG scans, indicating a potential role of PET-CT in monitoring, even in patients with negative serum CA-125. The cut-off value for CA-125 remains, however, a topic that is still debatable among studies.

The study performed by Evangelista et al. [35] also showed a higher predictive value for PET-CT compared to CA-125 for detecting recurrence. However, 35% of the cases with negative CA-125 were confirmed as positive through 18F-FDG PET-CT. Moreover, CA-125 was not correlated with the anatomical site of the lesions, and while high values of CA-125 were associated with peritoneal spread, they were not statistically significant.

Regarding ovarian cancer relapse, our study aligns with the results of the other studies mentioned here, thus indicating the potential role of PET-CT with 18F-FDG in evaluating this aspect of the disease.

The accuracy differences between PET-CT and serum CA-125 levels reported through multiple studies could also potentially justify the lack of correlation between positive PET-CT results and CA-125 in both recurrence and progressive disease in our patient group.

PET-CT scans and serum CA-125 levels could, however, be predictive factors in posttreatment overall survival [36]. Chu et al. observed that patients with both elevated CA-125 and positive PET-CT scans had the worst survival out of their study group, and those with both negative had the best prognosis. An intermediate survivability was observed at those that had just one of them positive.

As such, PET-CT might be useful both for the patients with elevated CA-125 levels (especially when conventional imaging is negative) and potentially also for those with negative values, since it may help predict their overall survival.

Therapeutic implications are of vital importance in the management of these patients. Imaging techniques, both conventional and nuclear medicine ones, along with serum marker changes, are the cornerstone of monitoring treatment efficiency. F18-FDG PET-CT led to a therapeutic change in 55.17% of the patients of our group, compared with only 17.24% after conventional imaging. Our results related to this aspect seem to correlate with the findings of other authors. Sousann et al. [37] reported more than a third of the cases (34%) being influenced in their decision making through the usage of PET-CT. A total of 24.7% of the patients from the study of Chung et al. [38] also had their therapeutical and monitoring strategy modified with the help of PET-CT. The importance of 18F-FDG PET-CT imaging in clinical decision making was also underlined by Simcock et al. [39], with the authors reporting a substantial proportion of the patients (58%) having their management of the disease altered after this imaging technique. As such, 18F-FDG PET-CT seems to have become a more relevant tool for assessing the efficiency of treatment in ovarian carcinoma, allowing for earlier changes in the therapeutic strategies. This underlines the crucial aspect of the metabolic changes of tumors that should be assessed alongside the morphological ones, with nuclear medicine imaging through 18F-FDG PET-CT remaining the only viable tool for achieving both at the same time at present.

The patients’ effective dose received in a chest/abdominal/pelvic CT may be 20–30 mSv [40], while at present, an 18F-FDG PET/CT scan, with a low-dose CT protocol, reaches only 5–12 mS. In this light, the results of our work confirm that the algorithm where 18F-FDG PET/CT is a first-line diagnostic method in staging and follow-up is adequate and benefits the patient, by reducing the radiation burden and allowing for prompt changes in the therapeutic scheme, if required [9].

### Limitations

The limitations of our study are related to the small number of patients. Moreover, the fact that this a retrospective study from a single institution might be a relative impediment in properly defining the role of PET-CT in the monitoring of ovarian carcinoma. A larger multicentric study might be more efficient in that regard.

The necessity to correlate conventional imaging techniques with PET-CT at the same relative value of CA-125 is another potential limitation, often the time frame in which we can compare them being restricted by that, since the patient can undergo only a limited number of procedures and examinations before another measurement of CA-125 or a therapeutic change.

Ovarian carcinoma is usually diagnosed at the moment of advanced disease. A total of 86.2% of our patients were already in stages III and IV at the moment of the diagnosis. Peritoneal involvement is also usually the norm, representing a difficult localization for the assessment of treatment response and metabolic and dimensional changes. Unless presented with the situation of “unequivocal progression”, the matter of progressive disease often remains a delicate diagnosis in this scenario. The clinical and biochemical data remain, as such, important for the correlation with the imaging techniques, in order to ensure the best outcome and management possible for these patients.

Additionally, because this is a retrospective study, the clinical and paraclinical information we had at our disposal was heterogenous. There are differences regarding the length of follow-up for each of the patients involved in the study and the moment of diagnosis, along with variable clinical, biochemical, histopathological and imaging data that were available to us.

## 5. Conclusions

In conclusion, PET/CT with 18F-FDG represents one of the most important imaging techniques used in the diagnosis and management of ovarian carcinoma. In our studied group, PET/CT with 18F-FDG was more useful than conventional imaging in detecting both recurrences of ovarian carcinoma and progressive disease. Peritoneal spread is usually the main route of disease progression, but lymphatic spread was also equally as significant in our group of patients.

Perhaps most importantly, F18-FDG PET-CT led to a therapeutic change in 55.17% of the patients in our group, compared with only 17.24% after conventional imaging. This underlines the crucial aspect of the metabolic changes of tumors that should be assessed alongside the morphological ones, with PET-CT remaining the only viable tool for achieving that at present.

## Figures and Tables

**Figure 1 healthcare-09-00666-f001:**
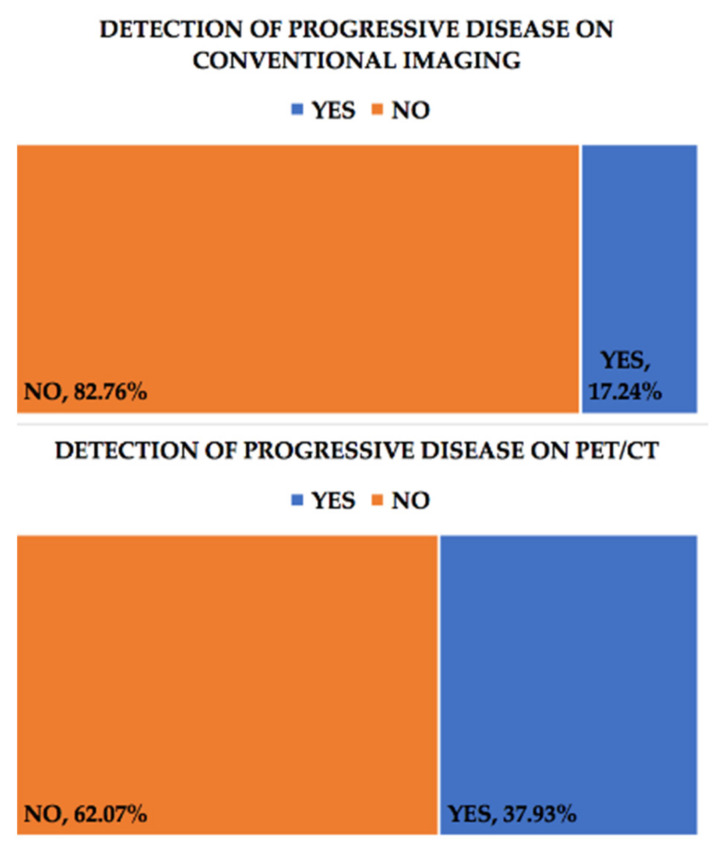
Progressive disease (PD) conventional imaging (upper chart) compared with F18-FDG PET-CT (lower chart).

**Figure 2 healthcare-09-00666-f002:**
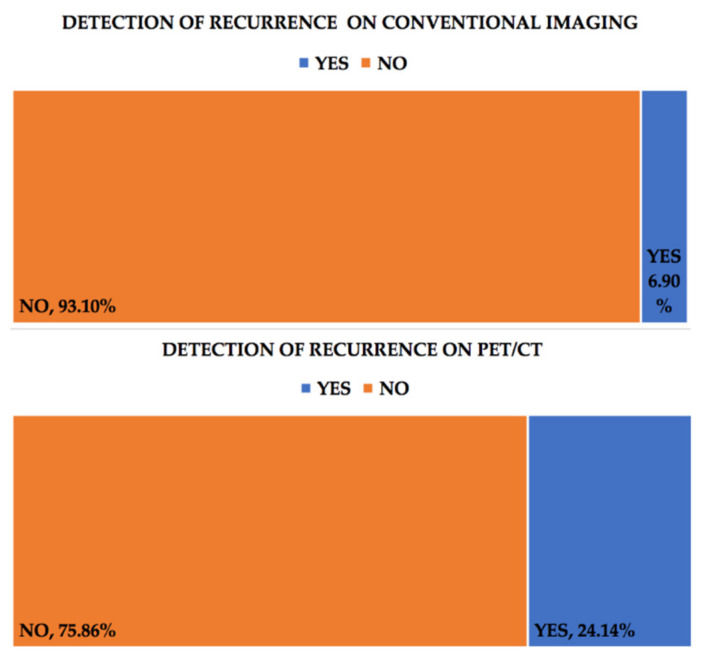
Recurrence of the disease in conventional imaging (upper chart) compared with F18-FDG PET-CT (lower chart).

**Figure 3 healthcare-09-00666-f003:**
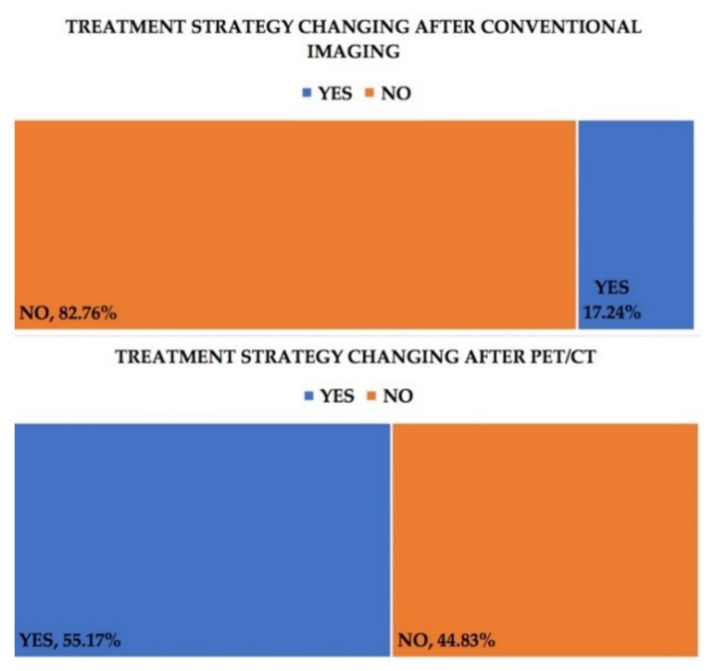
Treatment strategy change post-conventional imaging (upper chart) compared with F18-FDG PET-CT (lower chart).

**Figure 4 healthcare-09-00666-f004:**
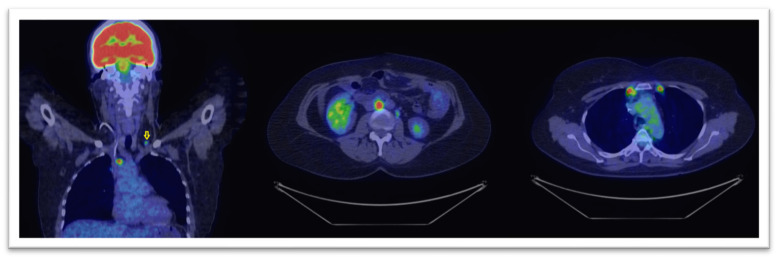
18F-FDG PET-CT scan with coronal (left image) and axial views (central and right) reveal intense tracer uptake in lumbo-aortic and mediastinal lymph nodes and also in a left supraclavicular one (marked with yellow arrow), in a patient with a CA-125 level of 280 U/mL. The lymphatic spread was suggestive for unexpected progressive disease.

**Figure 5 healthcare-09-00666-f005:**
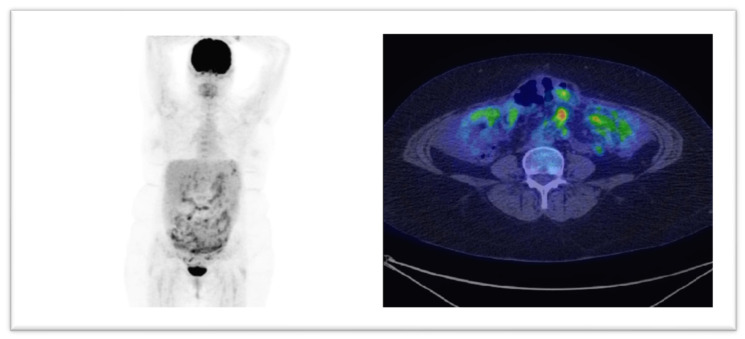
MIP and axial images from a 18F-FDG PET-CT scan, revealing peritoneal carcinomatosis, in patient with a CA-125 serum level of 642 U/mL. After the investigation, the type of chemotherapy was changed (paclitaxel to topotecan).

**Table 1 healthcare-09-00666-t001:** The patient group characteristics.

No.	Age Years	FIGO Staging	CA-125 (U/mL)	Progressive Disease on PET/CT	Progressive Disease on Conventional Imaging	Recurrence on PET/CT	Recurrence on Conventional Imaging
1	55	III	82.8	Yes	Yes	No	No
2	58	III	642	Yes	No	No	No
3	67	II	<35	No	No	No	No
4	78	NA	84.1	No	No	Yes	No
5	55	III	24.4	Yes	Yes	No	No
6	56	III	79	No	No	No	No
7	52	III	280	Yes	No	No	No
8	63	III	537	No	Yes	No	No
9	62	III	<35	No	No	No	No
10	50	III	102	Yes	Yes	No	No
11	59	III	1125	Yes	No	No	No
12	64	III	<35	No	No	Yes	No
13	60	III	1700	No	No	No	No
14	49	III	87.8	Yes	No	No	No
15	55	III	19.97	No	No	No	No
16	50	IV	126.3	No	No	Yes	No
17	47	I	23.1	No	No	No	No
18	72	III	463.7	Yes	Yes	No	No
19	64	III	37.35	No	No	No	No
20	60	III	53.22	Yes	Yes	No	No
21	54	III	737	Yes	No	No	No
22	51	III	1245	No	No	Yes	Yes
23	70	III	<35	No	No	No	No
24	60	NA	<35	No	No	No	No
25	51	III	85	No	No	Yes	No
26	65	III	<35 (10.4)	No	No	Yes	No
27	45	IV	<35 (14.51)	No	No	No	No
28	63	III	90.9	No	No	Yes	No
29	49	III	<35 (5.2)	Yes	No	No	No

FIGO—Federation of Gynecology and Obstetrics staging; NA—not available.

**Table 2 healthcare-09-00666-t002:** Sites of progressive disease found on F18-FDG PET-CT scans.

Site Where Disease Progression Was Observed through F-18 FDG PET/CT Imaging	Number of Cases/Percent Out of Total Number of Patients with Progressive Disease
Peritoneal	8 (72.7%)
Hepatic	4 (27.27%)
Splenic	2 (18.18%)
Subdiaphragmatic lymph nodes	5 (45.45%)
Supradiaphragmatic lymph nodes	3 (27.27%)

**Table 3 healthcare-09-00666-t003:** A representation of the treatment and patient management strategies being changed after the 18F-FDG PET-CT scan.

No.	Treatment before 18F-FDG PET-CT Scan	Treatment after 18F-FDG PET-CT Scan
1	Gemcitabine + Cisplatin	Cyclophosphamide + Cisplatin
2	Paclitaxel	Topotecan
3	Gemcitabine + Cisplatin	Cyclophosphamide + Cisplatin
4	Follow-up	Carboplatin + Paclitaxel
5	Olaparib	Trabectedin
6	Carboplatin + Paclitaxel	Carboplatin + Paclitaxel + Bevacizumab
7	Follow-up	Carboplatin + Paclitaxel
8	Follow-up	Carboplatin + Paclitaxel
9	Doxorubicin + Bevacizumab	Paclitaxel + Bevacizumab
10	Doxorubicin + Carboplatin	Topotecan
11	Carboplatin + Paclitaxel	Paclitaxel + Bevacizumab
12	Carboplatin + Paclitaxel	Carboplatin + Paclitaxel + Bevacizumab
13	Olaparib	Carboplatin + Paclitaxel
14	Gemcitabine	Surgery
15	Carboplatin + Paclitaxel	Gemcitabine + Carboplatin
16	Carboplatin + Paclitaxel	Doxorubicin + Bevacizumab

## Data Availability

Data supporting the reported results can be obtained, on request, from the corresponding author.

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
