# Peer review of "A Comparative Study between 18F-FDG PET/CT and Conventional Imaging in the Evaluation of Progressive Disease and Recurrence in Ovarian Carcinoma"

_healthcare, 2021, doi:10.3390/healthcare9060666_

Round 1

Reviewer 1 Report

The manuscript (healthcare-1192179) "A comparative study between 18F-FDG PET/CT and conventional imaging in the evaluation of progressive disease and recurrence in ovarian carcinoma" by George et al. had compared the efficiency of conventional imaging and hybrid imaging in detecting progressive disease and recurrences. The 18F-FDG-PET/CT hybrid imaging was also proven a valuable tool in assessing the prognosis of ovarian cancer. In this manuscript, the authors had tried to assess the impact of 18F-FDG PET-CT in the management of patients, specifically in their therapeutic approach. The manuscript is well written and the experiments were robust, the results were discussed adequately and supports their hypothesis. My specific comments are as follows- 

  1. In the material method section, all the patients enrolled in the study were clustered based on CA-125 ELISA (I believe), but no material methods were included for ELISA.
  2. The same was true for ultrasound imaging. As the authors have used ultrasound for their inclusion criteria, a brief comment on that would be helpful. 
  3. Figure 1 could be represented as a ROC curve and the change in prediction value could be represented more accurately. The authors have not discussed the "false positive/negative" cases that may cause the efficiency and specificity of this hybrid method to change. 
  4. If they decide to use ROC curve analysis for the sensitivity and specificity calculation, then it should be included in the material methods section. 
  5. The authors have discussed how the sensitivity increased over conventional imaging at each step of the disease. A visual representation would have been appealing to the reader.

the authors have discussed their findings in support of their hypothesis and also identified and acknowledged the limitations of this works as well. 

Author Response

Distinguish Reviewer 1:

Thank you very much for valuable comments made on our manuscript, which will improve significantly its quality.

Here are responses point-by-point and the specific corrections were made in the text of the manuscript.

“The manuscript (healthcare-1192179)

"A comparative study between 18F-FDG PET/CT and conventional imaging in the evaluation of progressive disease and recurrence in ovarian carcinoma" by George et al. had compared the efficiency of conventional imaging and hybrid imaging in detecting progressive disease and recurrences. The 18F-FDG-PET/CT hybrid imaging was also proven a valuable tool in assessing the prognosis of ovarian cancer. In this manuscript, the authors had tried to assess the impact of 18F-FDG PET-CT in the management of patients, specifically in their therapeutic approach. The manuscript is well written and the experiments were robust, the results were discussed adequately and supports their hypothesis. My specific comments are as follows- 

  1. In the material method section, all the patients enrolled in the study were clustered based on CA-125 ELISA (I believe), but no material methods were included for ELISA.
  2. The same was true for ultrasound imaging. As the authors have used ultrasound for their inclusion criteria, a brief comment on that would be helpful. 
  3. Figure 1 could be represented as a ROC curve and the change in prediction value could be represented more accurately. The authors have not discussed the "false positive/negative" cases that may cause the efficiency and specificity of this hybrid method to change. 
  4. If they decide to use ROC curve analysis for the sensitivity and specificity calculation, then it should be included in the material methods section. 
  5. The authors have discussed how the sensitivity increased over conventional imaging at each step of the disease. A visual representation would have been appealing to the reader.

the authors have discussed their findings in support of their hypothesis and also identified and acknowledged the limitations of this works as well”. 

Review 1:

  • We have added the following paragraph to the manuscript, briefly elaborating on the CA-125 sampling and measurement. The technique used was ECLIA.

“CA-125 was measured with electrochemiluminescence techniques (ECLIA). The CA-125 measurement was performed before administering each of the chemotherapy cycles and also during each follow-up control. The normal value of the laboratory is considered < 35 U/mL with a limit of detection of 0.6 U/mL. Venous blood was sampled, with a minimum volume of 0.5 mL of blood per sample. Sampling was performed under fasting conditions.

2) We have also briefly commented on the usage of ultrasound in the ‘Materials and Methods’ section (2.1).

3) As mentioned in the limitations of the study, we included a relatively small number of patients. Due to this, we were not certain that a ROC curve analysis would be entirely accurate, so we opted to not make use of one.

4) Regarding the possibility of false-positive and false-negative results, we have added in the ‘Materials and Methods’, section the comment that the PET-CT scans were performed after at least 2 months since the last chemotherapy cycle, in the case of the patients that were placed in follow-up evaluation. For patients undergoing chemotherapy, PET-CT done to monitor the treatment efficiency was performed either right before administering the next cycle or at least 2 weeks apart from the last one. In this way, one of the most important factors that could lead to false-positive results in oncological patients undergoing chemotherapy, the ‘flare’ effect, is reduced to a minimum.

Most of the other factors that can lead to false results are related to the preparation of the patient, proper functionality of the PET-CT system, the radiopharmaceutical etc. Some of them are described in the 2.2 section of the ‘Materials and Methods’ part, but most of them are common, routine practice and we didn’t feel it’s necessary to elaborate further.

If anything else requires additional clarification, please let us know. We express once more our gratitude towards you for taking the time to review our work and provide such an extensive feedback.

Reviewer 2 Report

My first comment is tat PET/CT is no longer referred to as hybrid imaging. I would like to know the chemotherapy agents used. Newer agents tend to produce better responses. Were there any patients treated with immunotherapy? I understand that therapy was changed in over 50% of patients. But, did it make a difference in overall survival? That is the most important question.  Was survival better than reported lengths of survival?

I am interested in GYN PET and there has not been any recent articles. I did not see any images. An example of how imges changed therapy would be helpful.

Words to change : imagisitic,, septaes. Also forego should be undergo

Author Response

Reviewer 2:

“My first comment is that PET/CT is no longer referred to as hybrid imaging. I would like to know the chemotherapy agents used. Newer agents tend to produce better responses. Were there any patients treated with immunotherapy? I understand that therapy was changed in over 50% of patients. But, did it make a difference in overall survival? That is the most important question.  Was survival better than reported lengths of survival?

I am interested in GYN PET and there has not been any recent articles. I did not see any images. An example of how imges changed therapy would be helpful.

Words to change :imagisitic,, septaes. Also forego should be undergo”

Review 2:

We would like to thank you first of all for the elaborate feedback, suggestions and for the effort invested in reviewing our article. We believe they have greatly helped us in improving the content of the article.

We will try to address the points that you have brought up, as it follows:

  • In the manuscript we have added a few 18F-FDG PET-CT images, which we consider to be illustrative for the discussed subjects, along with comments on each of them. Additionally, we have included a table which describes the therapeutic and patient management strategy changes after the PET-CT examination for each of the 16 patients where that occurred.
  • None of the patients in the group followed immunotherapy.
  • Regarding overall survival: The study was performed during a single year, during which we tracked the clinical evolution of the patients, but we have limited data regarding what happened afterwards, thus the evaluation of this aspect is not possible at the moment.
  • We have also removed the mention of ‘hybrid imaging’, rewording the syntax where it was appropriate. The mentioned words were corrected in the manuscript text.

Once more, we would like to express our gratitude towards the effort invested in reviewing our article and the pertinent suggestions that were brought up. If any other questions arise, please let us know.

Reviewer 3 Report

The authors performed a comparative study between FDG PET/CT and conventional imaging in the evaluation of recurrent or progressive ovarian carcinoma.

Unfortunately this article seems to not provide significant novelty compared to existing literature.

The advantages of FDG-PET/CT on conventional imaging for this indication are well known. Several original articles and evidence-based manuscripts have already documented the advantages of FDG-PET/CT in this setting. Furthermore, FDG-PET/CT is already included in international guidelines for the management of recurrent ovarian carcinoma.

Author Response

Review 3:

“The authors performed a comparative study between FDG PET/CT and conventional imaging in the evaluation of recurrent or progressive ovarian carcinoma.

Unfortunately this article seems to not provide significant novelty compared to existing literature.

The advantages of FDG-PET/CT on conventional imaging for this indication are well known. Several original articles and evidence-based manuscripts have already documented the advantages of FDG-PET/CT in this setting. Furthermore, FDG-PET/CT is already included in international guidelines for the management of recurrent ovarian carcinoma”.

We highly appreciate the commentaries and the time invested into reviewing our manuscript.

We would like to add that while we are aware that 18F-FDG PET-CT in ovarian carcinoma is already the subject of previous works, we desired to approach this subject in our paper for several reasons:

  • In our personal experience, while 18F-FDG PET-CT has certain well-documented advantages in other cancers, when compared with conventional imaging, the addressability towards the investigation in ovarian carcinoma is still not ideal. We find this particularly important, since ovarian carcinoma is usually diagnosed in advanced stages and the usage of PET-CT might significantly change the management of these patients.
  • Adding another instance of exposure towards 18F-FDG PET-CT felt useful to us, which is why we approached this subject, as other articles did before. A better approach based on personalized diagnostic protocol, limiting the exposure to ionizing radiations as much as possible, is one of the reason that we consider to submit for attention for Healthcare journal.

We understand however the point that is being brought up and we would like once more to express our gratitude for taking the time to review our work. If there are any other questions or suggestions, please let us know.

Round 2

Reviewer 3 Report

As reviewer, I have to state that the issue underlined in the first round of review still remains. The manuscript, even in the revised version, does not provide significant novelty.

Author Response

Distinguished reviewer, one again we express our gratitude for your comments and remarks. Please find below our response.

With respect, even in the latest version of the NCCN Guidelines Epithelial Ovarian Cancer/Fallopian Tube Cancer/ Primary Peritoneal Cancer Version 1.2021 (26 February 2021) for the staging, monitoring and recurrent disease the recommendation is to perform “chest/abdominal/pelvic CT, MRI, PET/CT, or PET (skull base to mid-thigh)”. The patients’ effective dose received in a chest/abdominal/pelvic CT  may be 20-30 mSv, while a low-dose PET/CT has 8-12 mSv.  In this light, the results of our work confirm that the place of PET/CT in the first line for staging and restaging, is not yet standardized. The FIGO and ESMO guidelines for ovarian cancers are from 2014, respectively 2013, where the PET/CT method is not even mentioned.

We insert some comments regrading your request, also in the manuscript.
